# Bayesian Layers: A Module for Neural Network Uncertainty

**Dustin Tran**
Google Brain

**Michael W. Dusenberry**
Google Brain*

**Mark van der Wilk**
Prowler.io

**Danijar Hafner**
Google Brain

## Abstract

We describe Bayesian Layers, a module designed for fast experimentation with neural network uncertainty. It extends neural network libraries with drop-in replacements for common layers. This enables composition via a unified abstraction over deterministic and stochastic functions and allows for scalability via the underlying system. These layers capture uncertainty over weights (Bayesian neural nets), pre-activation units (dropout), activations ("stochastic output layers"), or the function itself (Gaussian processes). They can also be reversible to propagate uncertainty from input to output. We include code examples for common architectures such as Bayesian LSTMs, deep GPs, and flow-based models. As demonstration, we fit a 5-billion parameter "Bayesian Transformer" on 512 TPUv2 cores for uncertainty in machine translation and a Bayesian dynamics model for model-based planning. Finally, we show how Bayesian Layers can be used within the Edward2 language for probabilistic programming with stochastic processes.[1]

```python
lstm = ed.layers.LSTMCellReparameterization(512)
output_layer = tf.keras.layers.Dense(10)

def loss_fn(features, labels, dataset_size):
  state = lstm.get_initial_state(features)
  nll = 0.
  for t in range(features.shape[1]):
    net, state = lstm(features[:, t], state)
    logits = output_layer(net)
    nll += tf.reduce_mean(
      tf.nn.softmax_cross_entropy_with_logits(
        labels[:, t], logits))

  kl = sum(lstm.losses) / dataset_size
  return nll + kl
```

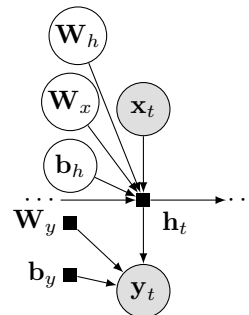

**Figure 1:** Bayesian RNN (Fortunato et al., 2017). Bayesian Layers integrates easily into existing workflows (here, a custom loss function followed by any training loop). Keras' `model.fit` is also supported. See Appendix A for comparisons to a vanilla TensorFlow, Edward1, and Pyro implementation.

**Figure 2:** Graphical model depiction. Default arguments specify learnable distributions over the LSTM's weights and biases; we apply a deterministic output layer.

# 1 Introduction

The rise of AI accelerators such as TPUs lets us utilize computation with $10^{16}$ FLOP/s and 4 TB of memory distributed across hundreds of processors (Jouppi et al., 2017). In principle, this lets us fit probabilistic models at many orders of magnitude larger than state of the art. We are particularly inspired by research on uncertainty-aware functions: priors and algorithms for Bayesian neural networks (e.g., Wen et al., 2018; Hafner et al., 2019), scaling up Gaussian processes (e.g., Salimbeni and Deisenroth, 2017; John and Hensman, 2018), and expressive distributions via invertible functions (e.g., Rezende and Mohamed, 2015).

Unfortunately, while research with uncertainty-aware functions are not limited by hardware, they are limited by software. Modern systems approach this by inventing a probabilistic programming language which encompasses all computable probability models as well as a universal inference engine (Goodman et al., 2012; Carpenter et al., 2016) or with composable inference (Tran et al., 2016; Bingham et al., 2019; Probtorch Developers, 2017). Alternatively, the software may use high-level abstractions in order to specify and fit specific model classes with a hand-derived algorithm (GPy, 2012; Vanhatalo et al., 2013; Matthews et al., 2017). These systems have all met success, but they tend to be monolothic in design. This prevents research flexibility such as utilizing low-level communication primitives to truly scale up models to billions of parameters, or in composability with the rich abstractions from neural network libraries.

Most recently, Edward2 provides lower-level flexibility by enabling arbitrary numerical ops with random variables (Tran et al., 2018). However, it remains unclear how to leverage random variables for uncertainty-aware functions. For example, current practices with Bayesian neural networks require explicit network computation and variable management (Tran et al., 2016) or require indirection by intercepting weight instantiations of a deterministic layer (Bingham et al., 2019). Both designs are inflexible for many real-world uses in research (see details in Section 1.1). In practice, researchers often use the lower numerical level—without a unified design for uncertainty-aware functions as there are for deterministic neural networks. This forces researchers to reimplement even basic methods such as Bayes by Backprop (Blundell et al., 2015)—let alone build on more complex baselines.

**Contributions.** This paper describes Bayesian Layers, an extension of neural network libraries which contributes one idea: instead of only deterministic functions as "layers", enable distributions over functions. Bayesian Layers does not invent a new language. It inherits neural network semantics to specify uncertainty models as a composition of layers. Each layer may capture uncertainty over weights (Bayesian neural nets), pre-activation units (dropout), activations ("stochastic output layers"), or the function itself (Gaussian processes). They can also be reversible layers that propagate uncertainty from input to output. Bayesian Layers can be used inside typical machine learning workflows (Figure 1) as well as inside a probabilistic programming language (Section 2.5).

To the best of our knowledge, Bayesian Layers is the first to: propose a unifying design across uncertainty-aware functions; design uncertainty as part of existing deep learning semantics; and demonstrate practical uncertainty examples on complex environments. We include code examples for common architectures such as Bayesian LSTMs, deep GPs, and flow-based models. We also fit a 5-billion parameter "Bayesian Transformer" on 512 TPUv2 cores for uncertainty in machine translation and a Bayesian dynamics model for model-based planning.

## 1.1 Related Work

There have been many software developments for distributions over functions. Our work takes classic inspiration from Radford Neal's software in 1995 to enable flexible modeling with both Bayesian neural nets and GPs (Neal, 1995). Modern software typically focuses on only one of these directions. For Bayesian neural nets, researchers have commonly coupled variational sampling in neural net layers (e.g., code from Gal and Ghahramani (2016); Louizos and Welling (2017)). For Gaussian processes, there have been significant developments in libraries (Rasmussen and Nickisch, 2010; GPy, 2012; Vanhatalo et al., 2013; Matthews et al., 2017; Al-Shedivat et al., 2017; Gardner et al., 2018), although flexible composability in the spirit of deep learning libraries remained a challenge.

Perhaps most similar to our work, Aboleth (Aboleth Developers, 2017) features variational BNNs and GPs. They uses a different design than Bayesian Layers, which we believe results in a less flexible framework that makes it more challenging to use for research. For example, their BNNs do not

support non-Gaussian priors or posterior approximations, different estimators, or probabilistic programming with a model-inference separation; their GPs only support random feature approximations; and they create a new neural network language instead of build on an existing one.

A closely related concept is MXFusion's probabilistic module (Dai et al., 2018), a module which implements a set of random variables alongside a dedicated inference algorithm. This has remarkable similarity to the way composing layers in Bayesian Layers ties estimation with the model specification (e.g., variational inference with deep GPs). Unlike MXFusion, Bayesian Layers enables a higher degree of compositionality to form the overall model, ultimately exploiting conditional independence relationships where, e.g., variational inference can be written as a series of layer-wise integral estimation problems. For example, deep GPs with variational inference in MXFusion involve a custom class whereas Bayesian Layers simply composes variational GP layers.

Another related concept is Pyro's random module (Bingham et al., 2019), a design pattern which lifts deterministic neural layers to Bayesian ones. This is done with effect handlers which replace weight instantiations with a Pyro primitive (typically `sample` on a distribution). Pyro's random module is effective for implementing Bayes by Backprop (Blundell et al., 2015), but it does not enable more recent estimators which avoid the high variance of weight sampling such as local reparameterization (Kingma et al., 2015), Flipout (Wen et al., 2018), and deterministic variational inference (Wu et al., 2018). More importantly, the random module focuses strictly on weight uncertainty whereas Bayesian Layers provides a unifying design across distribution over functions where uncertainty may exist anywhere in the computation—whether it be the weights, pre-activation units, activations, function, or propagating uncertainty from input to output.

Another related thread are probabilistic programming languages which build on the semantics of an existing functional programming language. Examples include HANSEI on OCaml, Church on Lisp, and Hakaru on Haskell (Kiselyov and Shan, 2009; Goodman et al., 2012; Narayanan et al., 2016). Neural network libraries can be thought of as a (fairly simple) functional programming language, with limited higher-order logic and a type system of (finite lists of) $n$-dimensional arrays. Similar to these works, Bayesian Layers augments the host language with methods for stochasticity.

## 2   Bayesian Layers

In neural network libraries, architectures decompose as a composition of "layer" objects as the core building block (Collobert et al., 2011; Al-Rfou et al., 2016; Jia et al., 2014; Chollet, 2016; Chen et al., 2015; Abadi et al., 2015; S. and N., 2016). These layers capture both the parameters and computation of a mathematical function into a programmable class.

In our work, we extend layers to capture "distributions over functions", which we describe as a layer with uncertainty about some state in its computation—be it uncertainty in the weights, pre-activation units, activations, or the entire function. Each sample from the distribution instantiates a different function, e.g., a layer with a different weight configuration.

### 2.1   Bayesian Neural Network Layers

The Bayesian extension of any deterministic layer is to place a prior distribution over its weights and biases. Bayesian neural networks have been to help address important challenges such as indicating model misfit (Dusenberry et al., 2019), generalization to out of distribution examples (Louizos and Welling, 2017), balancing exploration and exploitation in sequential decision-making (Hafner et al., 2019), and transferring knowledge across a collection of datasets (Nguyen et al., 2017). Bayesian neural net layers require several considerations. Figure 1 implements a Bayesian RNN; Appendix B implements a Bayesian CNN (ResNet-50).

**Computing the integral**   We need to compute often-intractable integrals over weights and biases $\theta$. Consider for example two cases, the variational objective for training and the approximate predictive distribution for testing,

$$\text{ELBO}(\theta) = \int q(\theta) \log p(\mathbf{y} \mid f_\theta(\mathbf{x})) \, \mathrm{d}\theta - \text{KL}\left[q(\theta) \,\|\, p(\theta)\right],$$

$$q(\mathbf{y} \mid \mathbf{x}) = \int q(\theta) p(\mathbf{y} \mid f_\theta(\mathbf{x})) \, \mathrm{d}\theta.$$

```python
class DenseReparameterization(tf.keras.layers.Dense):
  """Variational Bayesian dense layer."""
  def __init__(
    self,
    units,
    activation=None,
    use_bias=True,
    kernel_initializer='trainable_normal',
    bias_initializer='zero',
    kernel_regularizer='normal_kl_divergence',
    bias_regularizer=None,
    activity_regularizer=None,
    **kwargs):
    super(DenseReparameterization,
      self).__init__(..., **kwargs)
```

**Figure 3:** Bayesian layers are modularized to fit existing neural net semantics of initializers, regularizers, and layers as they deem fit. Here, a Bayesian layer with reparameterization (Kingma and Welling, 2014; Blundell et al., 2015) is the same as its deterministic implementation. The only change is the default for `kernel_{initializer,regularizer}`; no additional methods are added.

```python
if FLAGS.be_bayesian:
  Conv2D = ed.layers.Conv2DFlipout
else:
  Conv2D = tf.keras.layers.Conv2D

model = tf.keras.Sequential([
  Conv2D(32, 5, 1, padding='same'),
    tf.keras.layers.BatchNormalization(),
    tf.keras.layers.Activation('relu'),
    Conv2D(32, 5, 2, padding='same'),
    tf.keras.layers.BatchNormalization(),
    ...
])
```

**Figure 4:** Bayesian Layers are drop-in replacements for their deterministic counterparts.

Here, $\mathbf{x}$ may be a real-valued tensor as input features, $\mathbf{y}$ may be a vector-valued output for each data point, and the function $f$ encompasses the overall network as a composition of layers.

To enable different methods to estimate these integrals, we implement each estimator as its own Layer. The same Bayesian neural net can use entirely different computational graphs depending on the estimation (and therefore entirely different code). For example, sampling from $q(\theta)$ with reparameterization and running the deterministic layer computation is a generic way to evaluate layer-wise integrals (Kingma and Welling, 2014) and is used in Edward and Pyro. Alternatively, one could approximate the integral deterministically (Wu et al., 2018), and having the flexibility to vary the estimator as we do in Bayesian Layers is important when fitting the models in practice.

**Signature** We'd like to have the Bayesian extension of a deterministic layer retain its mandatory constructor arguments as well as its type signature of tensor-dimensional inputs and tensor-dimensional outputs. This enables compositionality, letting one easily combine deterministic and stochastic layers (Figure 4; Laumann and Shridhar (2018)). For example, a dense (feedforward) layer requires a `units` argument determining its output dimensionality; a convolutional layer also includes `kernel_size`.

**Distributions over parameters** To specify distributions, a natural idea is to overload the existing parameter initialization arguments in a Layer's constructor; in Keras, it is `kernel_initializer` and `bias_initializer`. These arguments are extended to accept callables that take metadata such as input shape and return a distribution over the parameter. Distribution initializers may carry trainable parameters, each with their own initializers.

For the distribution abstraction, we use Edward `RandomVariables` (Tran et al., 2018). Layers perform forward passes using deterministic ops and the `RandomVariables`. The default initializer represents a trainable approximate posterior in a variational inference scheme (Figure 3). By convention, it is a fully factorized normal distribution with a reasonable initialization scheme, but note Bayesian Layers supports arbitrarily flexible posterior approximations.[2]

**Distribution regularizers** The variational training objective requires the evaluation of a KL term, which penalizes deviations of the learned $q(\theta)$ from the prior $p(\theta)$. Similar to distribution initializers,

we overload the existing parameter regularization arguments in a layer's constructor; in Keras, it is `kernel_regularizer` and `bias_regularizer` (Figure 3). These arguments are extended to accept callables that take in the kernel or bias `RandomVariables` and return a scalar Tensor. By default, we use a KL divergence toward the standard normal distribution, which represents the penalty term common in variational Bayesian neural network training.

Importantly, note that Bayesian Layers does not have explicit notions for "prior" and "posterior". Instead, the layer reflects the actual computation within an algorithm and overloads existing semantics such as "initialization" (now the variational posterior) and "regularization" (now a KL divergence toward the prior). This is a tradeoff we made deliberately in that we lose separation of model and inference, but we benefit from the rich composability of network layers and integration with third-party libraries. (However, see Section 2.5 for how we might keep the separation if desired.)

## 2.2 Gaussian Process Layers

As opposed to representing distributions over functions through the weights, Gaussian processes represent distributions over functions by specifying the value of the function at different inputs. Recent advances have made Gaussian process inference computationally similar to Bayesian neural networks (Hensman et al., 2013). We only require a method to sample the function value at a new input, and evaluate KL regularizers. This allows GPs to be placed in the same framework as above.[3] Figure 5 implements a deep GP.

**Computing the integral**    Each Gaussian process prior in a model is represented as a separate Layer, which can be composed together. `GaussianProcess` implements exact (but expensive) conditioning. Approximations are given in the form of `SparseGaussianProcess` for inducing points (leading to Salimbeni and Deisenroth (2017)) and `RandomFourierFeatures` for finite trigonometric basis function approximations (used by Cutajar et al. (2017)). Both these approximations allow sampling from the predictive distribution of the function at particular inputs, which can be used for obtaining an unbiased estimate of the ELBO.

**Signature**    For the equivalent deterministic layer, maintain its mandatory arguments as well as tensor-dimensional inputs and outputs. For example, `units` in a Gaussian process layer determine the GP's output dimensionality, where `ed.layers.GaussianProcess(32)` is the Bayesian nonparametric extension of `tf.keras.layers.Dense(32)`. Instead of an `activation` function argument, GP layers have mean and covariance function arguments which default to the zero function and squared exponential kernel respectively. Any state in the layer's computational graph may be trainable such as kernel hyperparameters or inputs and outputs that the function conditions on.

**Distribution regularizers**    We use defaults which reflect each inference method's standard for training, e.g., no regularizer for exact GPs, a KL divergence regularizer on the inducing output distribution for sparse GPs, and a KL regularizer on weights for random projection approximations.

## 2.3 Stochastic Output Layers

In addition to uncertainty over the *mapping* defined by a layer, we may want to simply add stochasticity to the output. These outputs have a tractable distribution, and we often would like to access its properties: for example, auto-encoding with stochastic encoders and decoders (Figure 6); or a dynamics model whose network output is a discretized mixture density (Appendix C).[4]

**Signature**    To implement stochastic output layers, we perform deterministic computations given a tensor-dimensional input and return a `RandomVariable`. Because `RandomVariables` are Tensor-like objects, one can operate on them as if they were Tensors: composing stochastic output layers is valid. In addition, using such a layer as the last one in a network allows one to compute properties such as a network's entropy or likelihood given data.

Stochastic output layers typically don't have mandatory constructor arguments. An optional `units` argument determines its output dimensionality (operated on via a trainable linear projection); the default maintains the input shape and has no such projection.

```python
model = tf.keras.Sequential([
  tf.keras.layers.Flatten(),
  ed.layers.SparseGaussianProcess(
    units=256, num_inducing=512),
  ed.layers.SparseGaussianProcess(
    units=256, num_inducing=512),
  ed.layers.SparseGaussianProcess(
    units=10, num_inducing=512),
])
def loss_fn(features, labels):
  predictions = model(features)
  nll = tf.reduce_mean(
    tf.math.squared_difference(
      labels, predictions.mean()))
  kl = sum(model.losses)
  return nll + kl/dataset_size
```

**Figure 5:** Three-layer deep GP with variational inference (Salimbeni and Deisenroth, 2017; Damianou and Lawrence, 2013). We apply it for regression given batches of spatial inputs and vector-valued outputs. We flatten inputs to use the default squared exponential kernel; this naturally extends to pass in a more sophisticated kernel function.

```python
Conv2D = functools.partial(
  tf.keras.layers.Conv2D,
  padding='same', activation='relu')
Deconv2D = functools.partial(
  tf.keras.layers.Conv2DTranspose,
  padding='same', activation='relu')

encoder = tf.keras.Sequential([
  Conv2D(128, 5, 1),
  Conv2D(128, 5, 2),
  Conv2D(512, 7, 1, padding='valid'),
  ed.layers.Normal(name='latent_code'),
])
decoder = tf.keras.Sequential([
  Deconv2D(256, 7, 1, padding='valid'),
  Deconv2D(128, 5, 2),
  Deconv2D(128, 5, 1),
  Conv2D(3*256, 5, 1, activation=None),
  tf.keras.layers.Reshape([256, 256, 3, -1]),
  ed.layers.Categorical(name='image'),
])
def loss_fn(features):
  encoding = encoder(features)
  nll = -decoder(encoding).log_prob(features)
  kl = encoding.kl_divergence(
    ed.Normal(0., 1.))
  return tf.reduce_mean(nll + kl)
```

**Figure 6:** A variational auto-encoder for compressing 256x256x3 ImageNet into a 32x32x3 latent code. Stochastic output layers are a natural approach for specifying stochastic encoders and decoders, and utilizing their log-probability or KL divergence.

```python
model = tf.keras.Sequential([
  ed.layers.RealNVP(ed.layers.MADE([512, 512])),
  ed.layers.RealNVP(ed.layers.MADE([512, 512], order='right-to-left')),
  ed.layers.RealNVP(ed.layers.MADE([512, 512])),
])
def loss_fn(features):
  base = ed.Normal(loc=tf.zeros([batch_size, 32*32*3]), scale=1.)
  outputs = model(base)
  return -tf.reduce_sum(outputs.distribution.log_prob(features))
```

**Figure 7:** A flow-based model for image generation (Dinh et al., 2017).

## 2.4 Reversible Layers

With random variables in layers, one can naturally capture invertible neural networks which propagate uncertainty from input to output. This allows one to perform transformations of random variables, ranging from simple transformations such as for a log-normal distribution or high-dimensional transformations for flow-based models. We recommend using these layers when generative modeling with normalizing flows (Dinh et al., 2017) or understanding how networks make predictions (Jacobsen et al., 2018).

We make two considerations to design reversible layers:

**Inversion** Invertible neural networks are not possible with current libraries. A natural idea is to design a new abstraction for invertible functions such as TensorFlow's Bijectors (Dillon et al., 2017). Unfortunately, this prevents interoperability with existing layer and model abstractions. Instead, we simply overload the notion of a "layer" by adding an additional method `reverse` which performs the inverse computation of its call and optionally `log_det_jacobian`. A higher-order layer called `ed.layers.Reverse` takes a layer as input and returns another layer swapping the forward and reverse computation; by ducktyping, the reverse layer raises an error only during its call if `reverse`

```python
def model(input_shape):
  """Spatial point process."""
  rate = tf.keras.Sequential([
    ed.layers.GaussianProcess(64)
    ed.layers.GaussianProcess(input_shape)
    tf.keras.layers.Activation('softplus'),
  ])
  return ed.layers.PoissonProcess(rate)

def posterior():
  """Approximate posterior of rate function."""
  rate = tf.keras.Sequential([
    ed.layers.SparseGaussianProcess(
      units=64, num_inducing=512),
    ed.layers.SparseGaussianProcess(
      units=1, num_inducing=512),
    tf.keras.layers.Activation('softplus'),
  ])
  return rate
```

**Figure 8:** Cox process with a deep GP prior and a sparse GP posterior approximation. Unlike previous examples, using Bayesian Layers in a probabilistic programming language allows for a clean separation of model and inference, as well as more flexible inference algorithms.

is not implemented. Avoiding a new abstraction both simplifies usage and also makes reversible layers compatible with other higher-order layers such as `tf.keras.Sequential`, which returns a composition of a sequence of layers.

**Propagating Uncertainty** As with other deterministic layers, reversible layers take a tensor-dimensional input and return a tensor-dimensional output. In order to propagate uncertainty from input to output, reversible layers may also take a `RandomVariable` as input and return a transformed `RandomVariable` determined by its call, `reverse`, and `log_det_jacobian`.[5] Figure 7 implements RealNVP (Dinh et al., 2017), which is a reversible layer parameterized by another network (here, MADE (Germain et al., 2015)). These ideas also extend to reversible networks that enable backpropagation without storing intermediate activations in memory during the forward pass (Gomez et al., 2017).

## 2.5 Probabilistic Programming with Bayesian Layers

So far, the framework we laid out tightly integrates deep Bayesian modelling into existing ecosystems, but we have deliberately limited our scope. In particular, our layers tie the model specification to the inference algorithm (typically, variational inference). A core assumption for this to work is the modularization of inference per layer. This makes iterative procedures which depend on the full parameter space, such as Markov chain Monte Carlo, difficult to fit within the framework (but note, e.g., variational distributions with correlations across layers is possible because the layer integrals decompose conditionally).

Figure 8 shows that one can utilize Bayesian Layers in the Edward2 probabilistic programming language for more flexible modeling and inference. It does this by first specifying the prior generative process in the model program; any layers with approximations are moved into a separate program, the approximate posterior.[6] We could use, e.g., expectation propagation (Bui et al., 2016), which is possible with Edward2's tracing mechanism to manipulate the individual random variables within the model and posterior. Importantly, Bayesian Layers provides modeling semantics to enable arbitrary and scalable probabilistic programming in function space.

## 3 Experiments

We described a design for uncertainty models built on top of neural network libraries. In experiments, we aim to illustrate one point: Bayesian Layers is efficient and makes possible new model classes that haven't been tried before (in either scale or flexibility). The first experiment is machine translation, where training a model-parallel Bayesian model requires compatibility with Mesh TensorFlow's low-level communication operations. The second experiment is model-based reinforcement learning, where using a Bayesian dynamics model requires finetuning model updates across sequences of posterior actions using the TF Agents API (Guadarrama et al., 2018).

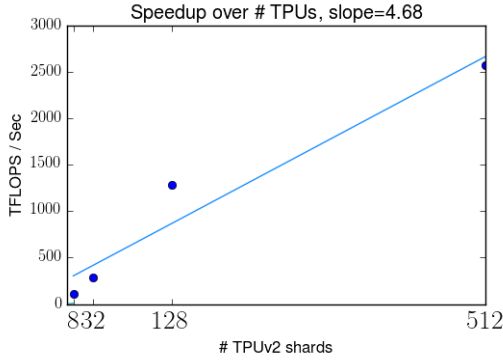

| | BLEU | Calibration Error |
|---|---|---|
| Baseline | **43.9** | 90.3% |
| Variational | **43.8** | **20.8%** |

**Figure 9:** Bayesian Transformer implemented with model parallelism ranging from 8 TPUv2 shards (core) to 512. As desired, the model's training performance scales linearly as the number of cores increases. It achieves the same BLEU score while also being well-calibrated.

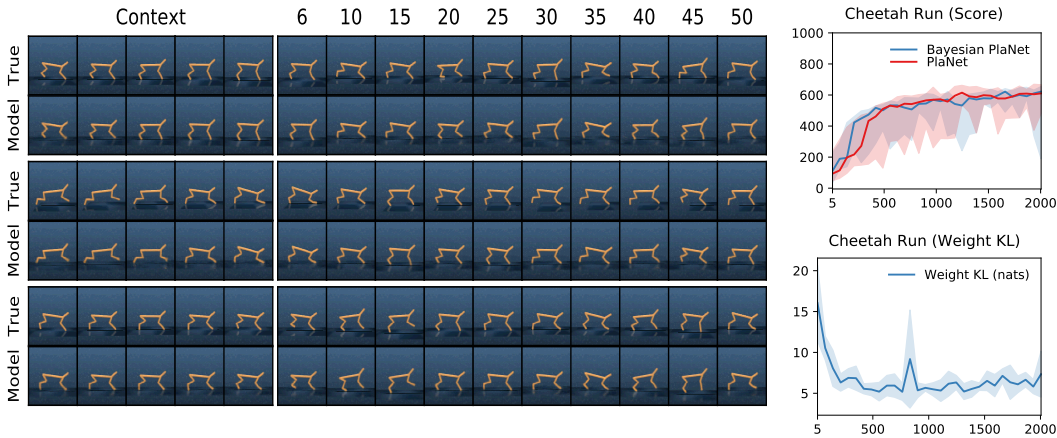

**Figure 10:** Results of the Bayesian PlaNet agent. The score shows the task median performance over 5 seeds and 10 episodes each, with percentiles 5 to 95 shaded. Our Bayesian version of the method reaches the same task performance. The graph of the weight KL shows that the weight posterior learns a non-trivial function. The open-loop video predictions show that the agent can accurately make predictions into the future for 50 time steps.

### 3.1 Model-Parallel Bayesian Transformer for Machine Translation

We implemented a "Bayesian Transformer" for the WMT14 EN-FR translation task. Using Mesh TensorFlow (Shazeer et al., 2018), we took a 2.8 billion parameter Transformer which reports a state-of-the-art BLEU score of 43.9. We then augmented the model by being Bayesian over the attention layers (using a stochastic layer with the Flipout estimator) and being Bayesian over the feedforward layers (using a stochastic layer with the local reparameterization estimator). Figure 9 shows that we can fit models with over 5-billion parameters (roughly twice as many due to a mean and standard deviation parameter), utilizing up to 2500 TFLOPs on 512 TPUv2 cores. Training time for the deterministic Transformer takes roughly 13 hours; the Bayesian Transformer takes 16 hours and 2 extra gb per TPU.

In attempting these scales, we were able to reach state-of-the-art BLEU scores while achieving lower calibration error according to the sequence-level calibration error metric (Kumar and Sarawagi, 2019). This suggests the Bayesian Transformer better accounts for predictive uncertainty given that the dataset is actually fairly small given the size of the model.

### 3.2 Bayesian Dynamics Model for Model-Based Reinforcement Learning

In reinforcement learning, uncertainty estimates can allow for directed exploration, safe exploration, and robust control. Still relatively few works leverage deep Bayesian models for control (Gal et al., 2016; Azizzadenesheli et al., 2018). We argue that this might be because implementing and train-

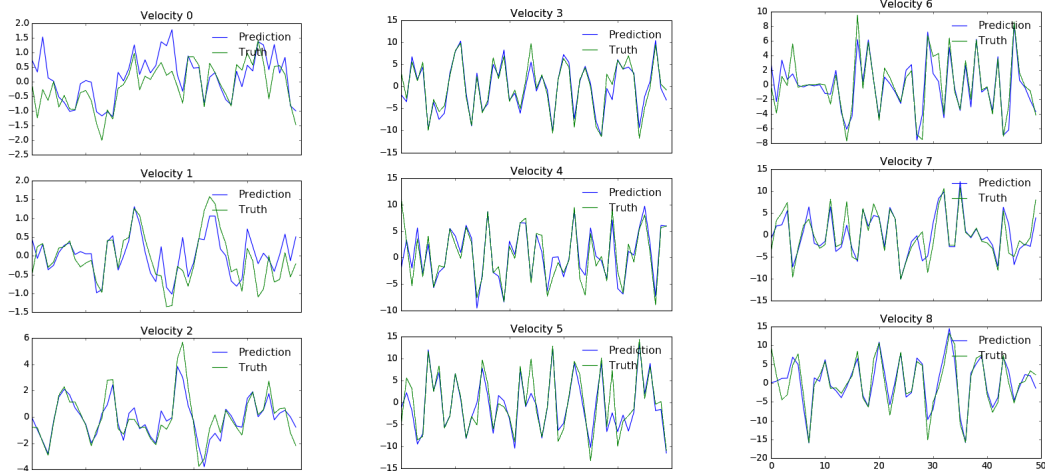

**Figure 11:** We use the Bayesian PlaNet agent to predict the true velocities of the reinforcement learning environment from its encoded latent states. Compared to Figure 7 of Hafner et al. (2018), Bayesian PlaNet appears to capture more information about the environment in the latent codes resulting in more precise velocity predictions.

ing these models can be difficult and time consuming. To demonstrate our module, we implement Bayesian PlaNet, based on the work of Hafner et al. (2018). The original PlaNet agent learns a latent dynamics model as a sequential VAE on image observations. A sample-based planner then searches for the most promising action sequence in the latent space of the model.

We extend this agent by changing the feedforward layers of the transition function to their Bayesian counterparts, `DenseReparameterization`. Bayesian PlaNet reaches a score of 614 on the cheetah task, matching the performance of the original agent (Figure 10). Training time for the deterministic dynamics model takes 20 hours, 8 gb; the Bayesian dynamics model takes 22 hours; 8 gb. We monitor the KL divergence of the weight posterior to verify that the model indeed learns a non-trivial belief. The result opens up many potential benefits for exploration and robust control; see Figure 11 for an example. It also demonstrates that incorporating uncertainty into agents can be straightforward given the right composability of software abstractions.

## 4 Discussion

We described Bayesian Layers, a module designed for fast experimentation with neural network uncertainty. By capturing uncertainty-aware functions, Bayesian Layers lets one naturally experiment with and scale up Bayesian neural networks, GPs, and flow-based models.

In future work, we are applying Bayesian Layers in our methodological and applied research, further expanding its support and examples. We are also exploring the use of uncertainty models in healthcare production systems, where the goal is to improve clinical decision-making by providing AI-guided clinical tools and diagnostics.

In Bayesian Layers, we encapsulated probabilistic notions as part of existing neural network abstractions such as layers, initializers, and regularizers. One question is whether this should also be done for other deep learning abstractions such as optimizers. Stochastic gradient MCMC easily fits on top of gradient-based optimizers by adding noise, as well as certain variational inference algorithms (Zhang et al., 2017; Khan et al., 2018). Further understanding this space, and how it interacts with probabilistic layers both in flexibility and inductive biases, is a potentially interesting direction.

## Footnotes

*Work done during Google AI residency.

[1]All code is available at https://github.com/google/edward2 as part of the `edward2` namespace. Code snippets assume `import edward2 as ed; import tensorflow as tf; tensorflow==2.0.0`.

[2] The only requirement for a distribution initializer is to return a sample (or most broadly, a Tensor of compatible shape and dtype). There is no restriction of independence across layers or tractable densities; hierarchical variational models (Ranganath et al., 2016) and implicit posteriors (Pawlowski et al., 2017) are compatible.

[3]More broadly, these ideas extend to stochastic processes. Figure 8 uses a Poisson process.

[4] In previous figures, we used loss functions such as `mean_squared_error`. With stochastic output layers, we can replace them with a layer returning the likelihood and calling `log_prob`.

[5]We implement `ed.layers.Discretize` this way in Appendix C. It takes a continuous `RandomVariable` as input and returns a transformed variable with probabilities integrated over bins.

[6]Above we used `GaussianProcess` for function priors. To specify function priors with Bayesian neural net layers, set the initializer to return the desired weight prior and remove the default regularizer.

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
