[Supplementary Material]

```
num_features = 128
units = 512

wx_loc = tf.Variable(tf.random.normal([num_features, 4 * units]))
wx_log_scale = tf.Variable(tf.random.normal([num_features, 4 * units]))
wh_loc = tf.Variable(tf.random.normal([units, 4 * units]))
wh_log_scale = tf.Variable(tf.random.normal([units, 4 * units]))
bias_loc = tf.Variable(tf.random.normal([4 * units]))
bias_log_scale = tf.Variable(tf.random.normal([4 * units]))
output_layer = tf.keras.layers.Dense(10)

def loss_fn(features, labels, dataset_size):
  qwx = wx_loc + tf.exp(wx_log_scale) * tf.random.normal([num_features, 4 * units])
  qwh = wh_loc + tf.exp(wh_log_scale) * tf.random.normal([units, 4 * units])
  bias = bias_loc + tf.exp(bias_log_scale) * tf.random.normal([4 * units])
  state = tf.ones(units)
  nll = 0.
  for t in range(features.shape[1]):
    z = tf.matmul(features[:, t], qwx) + tf.matmul(net, qwh) + bias
    i, f, o, u = tf.split(z, 4, axis=-1)
    i = tf.sigmoid(i)
    f = tf.sigmoid(f + 1.0)
    o = tf.sigmoid(o)
    u = tf.tanh(u)
    state = f * state + i * u
    net = output * tf.tanh(state)
    logits = output_layer(net)
    nll += tf.reduce_mean(tf.nn.softmax_cross_entropy_with_logits(labels[:, t], logits))

  kl = normal_kl_divergence(wx_loc, wx_log_scale, 0., 1.)
  kl += normal_kl_divergence(wh_loc, wh_log_scale, 0., 1.)
  kl += normal_kl_divergence(bias_loc, bias_log_scale, 0., 1.)
  kl /= dataset_size
  return nll + kl
```

**Figure 12:** Bayesian RNN in manual TensorFlow (Fortunato et al., 2017). Unlike Bayesian Layers, the LSTM must be manually written with explicit management over variables and the network computation. There is no encapsulation, for example, of computation given variational parameters and/or the prior.

## A    Bayesian RNN with vanilla TensorFlow and Edward

See Figure 12, Figure 13, and Figure 14.

## B    Bayesian ResNet-50

See Figure 15.

## C    Image Transformer with Stochastic Output Layers

See Figure 16.

## D    Vector-Quantized Variational Auto-Encoder

See Figure 17.

```python
def build_model(num_features, units):
  output_layer = tf.keras.layers.Dense(10)
  def model(features):
    qwx = ed.Normal(loc=tf.zeros([num_features, 4 * units]), scale=1., name="wx")
    qwh = ed.Normal(loc=tf.zeros([units, 4 * units]), scale=1., name="wh")
    bias = ed.Normal(loc=tf.zeros([4 * units]), scale=1., name="bias")
    state = tf.ones(units)
    logits = []
    for t in range(features.shape[1]):
      z = tf.matmul(features[:, t], qwx) + tf.matmul(net, qwh) + bias
      i, f, o, u = tf.split(z, 4, axis=-1)
      i = tf.sigmoid(i)
      f = tf.sigmoid(f + 1.0)
      o = tf.sigmoid(o)
      u = tf.tanh(u)
      state = f * state + i * u
      net = output * tf.tanh(state)
      logits.append(output_layer(net))
    return ed.Categorical(logits=logits)
  return model

def build_variational(num_features, units):
  wx_loc = tf.Variable(tf.random.normal([num_features, 4 * units]))
  wx_log_scale = tf.Variable(tf.random.normal([num_features, 4 * units]))
  wh_loc = tf.Variable(tf.random.normal([units, 4 * units]))
  wh_log_scale = tf.Variable(tf.random.normal([units, 4 * units]))
  bias_loc = tf.Variable(tf.random.normal([4 * units]))
  bias_log_scale = tf.Variable(tf.random.normal([4 * units]))
  def variational()
    qwx = ed.Normal(loc=wx_loc, scale=tf.exp(wx_log_scale), name="qwx")
    qwh = ed.Normal(loc=wh_loc, scale=tf.exp(wh_log_scale), name="qwh")
    qbias = ed.Normal(loc=bias_loc, scale=tf.exp(bias_log_scale), name="qbias")
    return qwx, qwh, qbias
  return variational

num_features = 128
units = 512
model = build_model(num_features, units)
variational = build_variational(num_features, units)

def loss_fn(features, labels, dataset_size):
  qwx, qwh, bias = variational()
  with ed.tape() as tape:
    with ed.condition(wx=qwx, wh=qwh, bias=qbias)):
      predictions = model(features)

  nll = -tf.reduce_mean(predictions.distribution.log_prob(labels))
  kl = qwx.distribution.kl_divergence(tape["wx"].distribution)
  kl += qwh.distribution.kl_divergence(tape["wh"].distribution)
  kl += qbias.distribution.kl_divergence(tape["bias"].distribution)
  kl /= dataset_size
  return nll + kl
```

**Figure 13:** Bayesian RNN in Edward2 (Tran et al., 2018), i.e., without using Bayesian Layers. The LSTM must be manually written with explicit management over variables and the network computation. While there is a clean separation of model and inference, much of the code is boilerplate and without encapsulation for a distribution over functions for composability.

```python
import pyro
import torch
import torch.nn as nn
from pyro.distributions import Categorical, Normal
from pyro.infer import SVI, Trace_ELBO
from pyro.optim import Adam

def standard_normal(*shape):
  return Normal(torch.zeros(*shape), torch.ones(*shape)).independent(1)

def build_model(num_features, units):
  lstm = nn.LSTM(num_features, units)
  output_layer = nn.Linear(units, 1)
  def model(features, labels):
    priors = {
        'weight_ih': standard_normal(4 * hidden_size, hidden_size),
        'weight_hh': standard_normal(4 * hidden_size, hidden_size),
        'bias_ih': standard_normal(4 * hidden_size),
        'bias_hh': standard_normal(4 * hidden_size),
    }
    lifted_module = pyro.random_module("lstm", lstm, priors)
    lifted_lstm = lifted_module()
    h_t = torch.zeros(batch_size, cell_size)
    c_t = torch.zeros(batch_size, cell_size)
    logits = []
    for feature_t in torch.chunk(features, sequence_length, dim=1):
      h_t, c_t = lifted_lstm(feature_t.squeeze(1), (h_t, c_t))
      logits.append(output_layer(h_t))
    logits = torch.cat(logits, dim=1)
    pyro.sample("obs", Categorical(logits=logits), obs=labels)
  return model

def variable_normal(name, *shape):
  loc = pyro.param(name + '_loc', torch.randn(*shape))
  scale = softplus(pyro.param(name + '_scale', torch.randn(*shape)))
  return Normal(loc, scale).independent(1)

def build_variational(num_features, units):
  def variational(input, target):
    dists = {
        'weight_ih': variable_normal('weight_ih', 4 * hidden_size, hidden_size),
        'weight_hh': variable_normal('weight_hh', 4 * hidden_size, hidden_size),
        'bias_ih': variable_normal('bias_ih', 4 * hidden_size),
        'bias_hh': variable_normal('bias_hh', 4 * hidden_size),
    }
    lifted_module = pyro.random_module('lstm', lstm, dists)
    return lifted_module()
  return variational

num_features = 128
units = 512
softplus = nn.Softplus()
model = build_model(num_features, units)
variational = build_variational(num_features, units)
inference = SVI(model, variational, Adam({'lr': 0.005}), loss=Trace_ELBO())
```

**Figure 14:** Bayesian RNN in Pyro (Bingham et al., 2019). It uses random module, which "lifts" variable instantiations in the built-in LSTM module and replaces them with a corresponding distribution. While there is a clean separation of model and inference, and encapsulation via the native LSTM module, Pyro uses an indirection which removes flexibility that is crucial to get Bayesian neural networks working in practice (e.g., lower variance estimators and priors which act on hidden units).

```python
def conv_block(inputs, kernel_size, filters, strides=(2, 2)):
  filters1, filters2, filters3 = filters
  x = ed.layers.Conv2DFlipout(filters1, (1, 1),
      strides=strides)(inputs)
  x = tf.keras.layers.BatchNormalization()(x)
  x = tf.keras.layers.Activation('relu')(x)
  x = ed.layers.Conv2DFlipout(filters2, kernel_size,
      padding='same')(x)
  x = tf.keras.layers.BatchNormalization()(x)
  x = tf.keras.layers.Activation('relu')(x)
  x = ed.layers.Conv2DFlipout(filters3, (1, 1))(x)
  x = tf.keras.layers.BatchNormalization()(x)
  shortcut = ed.layers.Conv2DFlipout(filters3, (1,1),
      strides=strides)(inputs)
  shortcut = tf.keras.layers.BatchNormalization()(shortcut)
  x = tf.keras.layers.add([x, shortcut])
  x = tf.keras.layers.Activation('relu')(x)
  return x

def identity_block(inputs, kernel_size, filters):
  filters1, filters2, filters3 = filters
  x = ed.layers.Conv2DFlipout(filters1,(1,1))(inputs)
  x = tf.keras.layers.BatchNormalization()(x)
  x = tf.keras.layers.Activation('relu')(x)
  x = ed.layers.Conv2DFlipout(filters2, kernel_size,
      padding='same')(x)
  x = tf.keras.layers.BatchNormalization()(x)
  x = tf.keras.layers.Activation('relu')(x)
  x = ed.layers.Conv2DFlipout(filters3, (1,1))(x)
  x = tf.keras.layers.BatchNormalization()(x)
  x = tf.keras.layers.add([x, inputs])
  x = tf.keras.layers.Activation('relu')(x)
  return x

def build_bayesian_resnet50(input_shape=None,
                            num_classes=1000):
  inputs = tf.keras.layers.Input(shape=input_shape,
                                 dtype='float32')
  x = tf.keras.layers.ZeroPadding2D((3, 3))(inputs)
  x = ed.layers.Conv2DFlipout(64, (7, 7),
      strides=(2, 2), padding='valid')(x)
  x = tf.keras.layers.BatchNormalization()(x)
  x = tf.keras.layers.Activation('relu')(x)
  x = tf.keras.layers.ZeroPadding2D((1, 1))(x)
  x = tf.keras.layers.MaxPooling2D((3,3), strides=(2,2))(x)
  x = conv_block(x, 3, [64, 64, 256], strides=(1, 1))
  x = identity_block(x, 3, [64, 64, 256])
  x = identity_block(x, 3, [64, 64, 256])
  x = conv_block(x, 3, [128, 128, 512])
  x = identity_block(x, 3, [128, 128, 512])
  x = identity_block(x, 3, [128, 128, 512])
  x = identity_block(x, 3, [128, 128, 512])
  x = conv_block(x, 3, [256, 256, 1024])
  x = identity_block(x, 3, [256, 256, 1024])
  x = identity_block(x, 3, [256, 256, 1024])
  x = identity_block(x, 3, [256, 256, 1024])
  x = identity_block(x, 3, [256, 256, 1024])
  x = identity_block(x, 3, [256, 256, 1024])
  x = conv_block(x, 3, [512, 512, 2048])
  x = identity_block(x, 3, [512, 512, 2048])
  x = identity_block(x, 3, [512, 512, 2048])
  x = tf.keras.layers.GlobalAveragePooling2D()(x)
  x = ed.layers.DenseVariationalDropout(num_classes)(x)
  model = models.Model(inputs, x, name='resnet50')
  return model

bayesian_resnet50 = build_bayesian_resnet50()
def loss_fn(features, labels):
  logits = bayesian_resnet50(features)
  nll = tf.losses.sparse_softmax_cross_entropy(
      labels=labels, logits=logits, reduction=tf.losses.reduction.MEAN)
  kl = sum(bayesian_resnet50.losses) / dataset_size  # KL are Layer side-effects
  return nll + kl

# Alternatively, run the following instead of a custom training loop.
model.compile(optimizer=tf.train.AdamOptimizer(),
              loss='categorical_crossentropy',
              metrics=['accuracy'])
model.fit(features, labels, batch_size=32, epochs=5)
```

**Figure 15:** Bayesian ResNet-50.

```python
def build_image_transformer(hparams):
  x = tf.keras.layers.Input(shape=input_shape)
  x = ChannelEmbedding(hparams.hidden_size)(x)
  x = tf.pad(x, [[0, 0], [1, 0], [0, 0]])[:, :-1, :])
  x = PositionalEmbedding(max_length=128*128*3)(x)
  x = tf.keras.layers.Dropout(0.3)(x)
  for _ in range(hparams.num_layers):
    y = MaskedLocalAttention1D(hparams)(x)
    x = LayerNormalization()(
      tf.keras.layers.Dropout(0.3)(y) + x)
    y = tf.keras.layers.Dense(
      x, hparams.filter_size, activation='relu')
    y = tf.keras.layers.Dense(
      hparams.hidden_size, activation=None)(y)
    x = LayerNormalization()(
      tf.keras.layers.Dropout(0.3)(y) + x)
  x = ed.layers.MixtureLogistic(3, num_components=5)(x)
  outputs = ed.layers.Discretize(x)
  model = tf.keras.Model(inputs, outputs, name='ImageTransformer')
  return model

transformer = build_image_transformer(hparams)
def loss_fn(features):
  return -tf.reduce_mean(transformer(features).distribution.log_prob(features))
```

**Figure 16:** Image Transformer with discretized logistic mixture (Parmar et al., 2018) over 128x128x3 features. Stochastic output layers let one easily experiment with the likelihood. We assume layers which don't exist in Keras; functional versions are available in Tensor2Tensor (Vaswani et al., 2018).

```python
base_depth = 128

encoder = tf.keras.Sequential([
    tf.keras.layers.Conv2D(base_depth, 5, 1, padding='same', activation='relu'),
    tf.keras.layers.Conv2D(base_depth, 5, 2, padding='same', activation='relu'),
    tf.keras.layers.Conv2D(2 * base_depth, 5, 1, padding='same', activation='relu'),
    tf.keras.layers.Conv2D(2 * base_depth, 5, 2, padding='same', activation='relu'),
    tf.keras.layers.Conv2D(4 * latent_size, 7, padding='valid', activation='relu'),
    tf.keras.layers.Flatten(),
    ed.layers.VectorQuantizer(512, name='latent_code'),
])
decoder = tf.keras.Sequential([
    tf.keras.layers.Conv2DTranspose(2 * base_depth, 7, padding='valid', activation='relu'),
    tf.keras.layers.Conv2DTranspose(2 * base_depth, 5, padding='same', activation='relu'),
    tf.keras.layers.Conv2DTranspose(2 * base_depth, 5, 2, padding='same', activation='relu'),
    tf.keras.layers.Conv2DTranspose(base_depth, 5, padding='same', activation='relu'),
    tf.keras.layers.Conv2DTranspose(base_depth, 5, 2, padding='same', activation='relu'),
    tf.keras.layers.Conv2DTranspose(base_depth, 5, padding='same', activation='relu'),
    tf.keras.layers.Conv2D(3, 5, padding='same', activation=None),
    ed.layers.Bernoulli(256*256*3, name='image'),  # likelihood
])
def loss_fn(features):
  encoded_features = encoder(features)
  reconstruction = -decoder(encoded_features).distribution.log_prob(features)
  penalties = sum(encoder.losses)
  return tf.reduce_mean(reconstruction + penalties)
```

**Figure 17:** Vector-quantized variational auto-encoder (van den Oord et al., 2017) for 256x256 ImageNet. VQVAEs assume a uniform prior during training; we use a deterministic encoder.