[Reviews · NeurIPS 2019]

Reviewer 1



EDIT post-rebuttal: Thanks to the authors for answering to my comments. I am still voting for acceptance of this paper. +++++ This paper is about a software component, called Bayesian Layers, that allows for consistent creation of deep layers that are associated with some form of uncertainty or stochasticity. The paper outlines the design philosophy and principles, shows many examples and concludes with new demonstrations of Bayesian neural network applications. I find that this work is on a significant topic, since software for Bayesian (deep) learning models significantly lacks behind. Integration and drop-in replacement with traditional architectures seems like the right avenue to pursue, and is a strong motivation point for this approach. I also think that this work is sufficiently original, related to what one could expect form a software component. I find that the relation to Pyro's random module is strong, perhaps stronger than discussed in the paper, when it comes to the fundamental concepts behind it (for this statement I am assuming that there is no fundamental reason why Pyro couldn't be extended with recent estimators using the current random module's architecture, but I am not an expert). When it comes to practicality though, I find that Bayesian Layers is sufficiently different, more consistent and more extensive. In any case, the community needs more works on this kind of modules. Having said that, comparison to Pyro (e.g. more discussion or side-by-side snippets) would be useful. Regarding quality, it is in general difficult to criticize a software in the same way as for a mathematical idea: ultimately, in software it's all about trade-offs in functionality. Nevertheless, I find that from the technical viewpoint, the design choices are very reasonable, offer useful features and improve upon past approaches (e.g. Pyro's random module) in certain aspects. The consistent use of layers with traditional architectures and software components makes the approach very useful, readable and scalable. The separation of model and inference is an important aspect for the Bayesian community, and is indeed unfortunate that the default use of Bayesian Layers obscures this separation. However, it is also true that no one design can fulfill all requirements, and the authors admit that this is deliberate to facilitate other important aspects of their framework. Indeed the modularization of inference per layers is a very interesting idea, however for such an important design choice, I feel that the paper does not explain clearly all the reasons why this is beneficial. A more concrete side-by-side demonstration about its advantages would be welcome. As a side note, it is also not entirely clear how the approach in section 2.5 separates in principle model and inference. Could the authors provide a more extended snippet that uses the model and the posterior? As for presentation, in this type of software paper the usual issue of trading-off details and abstraction is exacerbated. However, I feel that the authors did a good job exposing the correct amount of details. Having said that, the flow between being high- and low-level is not very smooth and at times the paper assumes too deep knowledge of Keras, Tensorflow and recent Bayesian ML methods (e.g. lines 22-40). The intersection of readers being in-depth experts in all the above is smaller than the set of readers in the target audience. Therefore, I suggest a background section listing the basic topics before diving deeper into the details. The authors use a variety of ML models to demonstrate the concepts. Although this is more difficult to follow, I feel that it is ultimately more instructive, and also demonstrates better the range of applications for Bayesian Layers. Related work is generally discussed thoroughly. Overall, the presentation in this paper is honest (starting with what it is: code), and gets right to the point. A think that the paper would benefit from the following central discussion: Bayesian models are still not as widely used in practice as DNNs. Some say that this is due to current software capabilities. Is Bayesian Layers a reply to this? In my opinion, it seems that there's more missing, whether in the inference methods, practical tricks or software. It'd be interesting to see a discussion about this here. The paper shows *how* to use Bayesian layers, but not as much *why*. This is more important than in theoretical papers, because by (some) definition a software paper is about practicality. More than simply a discussion on the above, it'd be nice to have a demonstration, e.g. with stronger, previously unthought of RL results. However, that would be an impressive bonus, and I understand that it goes beyond the scope of the paper.

Reviewer 2



-- Paper Summary -- This paper describes a comprehensive extension to Edward2, built upon Tensorflow, which permits the seamless inclusion of non-deterministic layers for constructing deep probabilistic models. The API is similar to that used for standard neural network layers, but permits the inclusion of uncertainty on a variety of components such as the weights in a layer, any associated parameters in the activation function, as well as the inclusion of Gaussian process layers (for which a variety of formulations are included). This module not only makes it easier for practitioners to construct such probabilistic models, but also exploits the processing power of state-of-the-art TPUs for implementing large-scale models such as the Bayesian transformer described in the paper. -- Writing/Clarity -- The paper is well-written and methodological in its approach. The related work is properly described, and I found the individual subsection on different layer options to be informative and useful. On the downside, I found the code snippets to be fairly messy - the presentation is quite poor with some instances of overlapping text, and I highly encourage the authors to rethink their presentation. I am also neither a fan of the current side-by-side placement of the figures which is hard to follow at times. With particular emphasis on the paper title and the first line of the abstract, I am slightly puzzled by the insistence on referring to the module as catering for ‘neural networks’ when it also enables the inclusion of Gaussian process layers, and consequently the construction of deep Gaussian processes. Of course we could debate on the connection between Gaussian processes and infinitely-wide neural networks, but this generalisation appears misleading to me in this context (and possibly also undersells the broader functionally provided by the proposed module). It may also be helpful to include a more detailed breakdown of the various options available for some of the arguments in the method signatures. For example, the local reparameterisation trick is mentioned in the description of one of the experimental set-ups, but it would be nice to have a more comprehensive list of the available options for each parameter option (perhaps in the appendix). I appreciate that this verges on providing full code documentation, but I think it would be nice to have more of this included in the paper along with a list of associated references for the papers originally proposing the implemented/featured techniques. Some additional minor complaints: - There are some typing issues in the references where words such as ‘Gaussian’ appear as lower-case. I noted some papers cited here as appearing on ‘arXiv’ which have since been published at either ICLR or ICML 2019 - please double check and update accordingly. - A few typos/preferences: L29: monolothic -> monolithic; L32: ops -> operations; L85: abbreviated last names in citation; L207: In ‘the’ experiments; - The phrase ‘whether it be the weights, pre-activation units, activations’ appears in some form or another at least 3 times in the first two pages. While I appreciate the authors’ intent to drive the message home, it unintentionally comes across as overly repetitive. -- Originality and Significance -- The importance of Bayesian inference in practical applications of machine learning has recently become more prevalent, and the work presented here is certainly a step forward in facilitating their use. As the authors clearly illustrate in the supplementary material, implementing Bayesian variations of neural networks and other models using frameworks such as Tensorflow and PyTorch typically require a substantial amount of tweaking (often bordering on ‘hacks’), whereas the proposed module could help abstract away from such complex implementations. Homogenising techniques such as Bayesian neural networks and Gaussian processes under a single framework should also have positive repercussions by encouraging future work on these topics to include broader experimental comparisons of both methods. While reading the paper, I was wondering about the possibility of incorporating the recent work by Sun et al. 2019 in this set-up, so I was very pleased to see this listed as a direction for future work. I also liked that the authors picked reinforcement learning as a use-case for demonstrating the scope of this work. Given the ballooning popularity of this field, showing how Bayesian Layers enable the extension of these methods to the Bayesian setting should be a great entry point for both ‘Bayesians’ interested in dipping their toes into reinforcement learning, and vice versa. The related work section is comprehensive and I appreciated the segment dedicated to describing the differences to Pyro, which would be the primary competing mechanism available on PyTorch. However, I was surprised that there was no mention of the MXFusion package (Dai et al., 2018), which completes the trifecta of modular probabilistic modelling by offering an implementation for MXNet. To the best of my understanding, this package bears greater similarity to the work presented here due to the inclusion of Gaussian processes and a similar notion of ‘inference modularity’ relying on variational inference. I would expect to see any connections explored in greater detail given the similar nature of this work. Instead, it is currently conspicuous by its absence. Work by Cutajar et al (2017) on deep Gaussian processes also merits a mention in the discussion on GPs for being one of the first practical instances of exploiting Tensorflow for implementing large-scale DGPs. -- Technical Quality/Evaluation -- There are little theoretical elements to comment on here, but I found most of the discussion on the implementation details of this module sufficiently clear to follow. Perhaps including some more background information on Edward2 could be useful for readers who aren’t immediately familiar with what it currently provides. As highlighted earlier in my review, the presentation and content of the code snippets could definitely be improved however. I think it’s also important to specify the degree to which this work extends beyond simply building an API around existing functionality and implemented models - at the moment the extent of the contributions are not entirely clear. Although the experiments showcasing the module are diverse and sufficiently convey the broad scope of their potential use, I feel as though the paper is missing some degree of benchmarking with regards to both scalability across hardware and model parallelisation. While I appreciate that this is not immediately within the scope of this work (which rather relies on the fundamentals of Tensorflow for these inner workings), it would be interesting to see whether a direct comparison against Pyro and MXFusion can be carried out in this regard. In its current format, the experimental evaluation feels fairly isolated in simply showcasing the functionality of the module, but there is little external context. It is also slightly unclear to me how existing frameworks such as GPflow are positioned in relation to this module - while I appreciate that the scope of GPflow is much greater than the brief appearance of GPs featured here, I am still curious to understand whether say, the DGP of Salimbeni et al. (2017) which can be constructed here, is just as good as the original implementation. -- Overall recommendation -- I would be hard-pressed to classify this paper as ‘essential reading’, but I also believe that it successfully describes the module in a succinct manner, while also giving potential readers and conference attendees a better incentive for incorporating probabilistic modelling in their workflows. There’s a few disappointing aspects which I highlighted in the review - aside from some other minor issues, the messy inclusion of code snippets is a sign of carelessness when preparing the submission. While effective in showcasing the diversity of model which can be tackled using the proposed module, the experimental section also verges on being more of a ‘demo’ than a critical evaluation. I am currently giving this submission a relatively ‘modest’ score, but would be keen on raising this score following a convincing rebuttal. ** Post-rebuttal update ** Thank you for your response! The rebuttal targets the majority of concerns listed in the reviews, and also clears up some of the more muddled aspects of the paper. If accepted, there are some points which require particular attention, especially clarifying the similarities to related work (all reviewers had issues with this aspect) and highlighting the contributions further. Papers describing software toolkits are always faced with greater scepticism, which makes it essential to clearly emphasise the contributions of the paper. Likewise, the inclusion of additional comparisons and benchmarks will elevate this from seeming like a standard technical report or documentation to a proper paper. Coupled with the presentation issues highlighted across reviews, I believe there is still some work to do, which is why I am not increasing my score. However, I also think that this work could benefit from the increased attention enabled by NeurIPS, and hopefully encourage more streamlined model implementation and evaluation within the Bayesian community. For this reason, my vote still tends towards accepting this paper. * With regards to the title, it is ultimately at your discretion whether to change it or not. However, ‘uncertainty-aware functions’ has a nice ring to it!

Reviewer 3



---------------------------------------------------------------------------------------------------------- Post-rebuttal update: ================ I thank the authors for the clarification. After some discussions with the other reviewers and the AC, I've decided to increase my score to lean more towards an acceptance. I do believe, however, the current version of the paper is sub-par in term of presentation. Please add comparison with Aboleth, along with other missing information that I had mentioned in my original review, and please fix all the formatting issues. I do hope that given sufficient work from the authors in preparing the camera-ready version, this paper could be more like a proper scientific paper, instead of a description of a software toolkit. ---------------------------------------------------------------------------------------------------------- This article describes an extension of TensorFlow (TF) called "Bayesian Layers" (BL) which abstracts the variational inference implementations (e.g. sampling, reparametrization trick, and KL-divergence calculation) inside the API itself. These layers are constructed in such a way that they maintain the compatibility with the pre-existing API in TF. The resulting layers are therefore maintained compatibility with the pre-existing layers in TF. This allows users to stack together variational and vanilla layers together when building their models. Some examples provided by the authors include the variational versions of fully-connected, convolutional, RNN, and Gaussian process layers. The article shows that these layers can be used as drop-in replacements for the existing deterministic layers in (possibly any) existing models to enable uncertainty quantification, which is very important in real-world systems. Furthermore, the authors demonstrate that the proposed layers scale well to a particularly large model with 5 billion parameters. I like the idea of having an easy way of building BNNs. Especially, having a painless way to turn complex, deterministic models (like what people use in NLP or CV) to Bayesian ones is very appealing. Even more so because the real-world systems nowadays cannot quantify their uncertainty, giving rise to many safety issues. So, I think there is a big real-world potential for this toolkit. Having said that, I have the feeling that the proposed toolkit is a bit too similar to the existing toolkits such as Aboleth (https://github.com/gradientinstitute/aboleth). Indeed, the authors cite Aboleth as the most similar toolkit to theirs. However, there is a lack of comparison between BL and Aboleth, thus it is difficult to know what makes BL special and original. Furthermore, when comparing BL with Pyro, the authors mentioned that BL could use more recent estimators such as Flipout and deterministic VI. I think this could be a strong point for BL, but the authors did not do any follow-up on this feature. While BL hides away the pain of implementing VI, it makes the toolkit inflexible. For example, I am under the impression that only VI with Gaussian prior and variational posterior (with diagonal covariance) is supported. While Edward can be used on top of BL for more advanced inference, it is still not clear to me how easy this would be and whether the usage of BL with non-Gaussian priors and posteriors could be done easily. Perhaps the author could clarify further on this point. For the experiments, I think it is really great to see that one can use the proposed toolkit to turn a large, complex model into a Bayesian one. But, I think an additional comparison with the respective vanilla deterministic model in term of training time and memory overhead would be important. I also think the claim in Figure 9 that the performance of BL scales linearly is not warranted as there are not enough data points to draw that conclusion. That is, there is a lot of uncertainty on how does the performance curve look like between x=128 and x=512. The overall writing of the article is clear, although some questionable terms such as "tensor-dimensional" are being used here and there. I appreciate the authors for showing codes describing the usage of BL. However, the presentation of those codes could be better, as they often overlap with each other and cross the page boundary. Another minor point that I would like to bring up is that in the x-axis of Figure 9, the number 8 and 32 are too close together and makes it confusing at a glance. Finally, I would like to mention again that I really like what the authors proposed in this article and hope that it could have a big impact on real-world systems. However, ultimately, I think this article's scientific significance is low, as the nature of this article is a description of a software toolkit.

[Author Response · NeurIPS 2019]

Thanks for the comments! Note the reviewers all found Bayesian Layers (BL) to have significant practical impact. R2
and R3 also found the design to be novel, while R4's major criticism is that the design is too similar to Aboleth's. (We
hope to convince R4 that this is not true below). Below we answer major comments; we'll fix the minor ones.

**REVIEWER 2** **Pyro's random module** Pyro can't be extended with recent estimators. It can only change how
weights are instantiated in deterministic layers. To change the computation itself requires a design similar to BL's.

**"When it comes to practicality, I find that BL is sufficiently different, more consistent and more exten-**
**sive...Having said that, comparison to Pyro would be useful."** Thanks! In our revision, we'll include a Pyro
implementation of Figure 1 (the Bayesian RNN). Note Appendix A includes implementations in raw TF and Edward.

**extended snippet separating model and inference** We'll link to an end-to-end script and which involves real data.

**"I suggest a background section listing the basic topics before diving deeper into the details."** That's a good idea!
We'll clean up the presentation and start with basic topics as background.

**larger discussion about limitations of Bayesian models to DNNs** Great idea. We'll add to the discussion. BL is
indeed a reply to the software limitation. More open limitations exist: targeted applications; practical tricks; baselines.

**REVIEWER 3** **"I am slightly puzzled by the insistence on referring to the module as catering for 'neural**
**networks'."** We can revise the title to "functional uncertainty" or "uncertainty-aware functions." Let us know!

**MXFusion** We're happy to mention MXFusion. Studying the paper and code, MXFusion's "probabilistic module"
is highly related in tying models with an inference algorithm. Unfortunately, I'm not clear on how their inference
modularity composes. For example, their support for deep GPs is preliminary. Their GitHub branch (**url**) suggests that
this involves a custom class rather than BL's approach of simply composing variational GP layers.

**Cutajar et al (2017)** It's indeed relevant! We'll cite it.

**clearer summary of contributions** We'll add a Contributions section. In summary, BL is the first to: contribute a
unifying design across uncertainty-aware functions (BNNs, GPs, stochastic outputs, reversible layers); enable a BNN/GP
design as part of existing semantics+ecosystems; and demonstrate practical examples on complex environments.

**more benchmarks.** Our current experiments apply BNNs with features not present in previous libraries. We decided
not to benchmark against Pyro and MXFusion because this would reduce to a benchmark of their backends. For the
Bayesian Transformer, we have benchmarked Examples/Sec under libraries with the same backend however (i.e., vanilla
TF and Edward). In both graph and eager mode, runtimes are the same with negligible differences from simulation
variance; we'll include more details in the paper. See also the comment to R4 about additional comparisons.

**"I am still curious to understand whether say, the DGP of Salimbeni et al. (2017) which can be constructed here,**
**is just as good as the original implementation."** It's mathematically the same as Salimbeni et al. (2017). Note this is
not the same, however, as Damianou and Lawrence (2013). The latter may be what MXFusion is implementing by
constructing a separate (not composable) class for DGPs; BL's approach simply composes variational GP layers.

**REVIEWER 4** **Aboleth.** Aboleth uses a different design, ending up with a less flexible framework that makes it
more challenging to use for research. We'll add the following points to the revision:
• **For BNNs, Aboleth hardcodes Gaussian priors and Gaussian posteriors.** BL supports arbitrary priors and
posteriors, including implicit distributions and hypernetworks. BL also supports arbitrary estimators, including local
reparameterization and deterministic VI (Sec 2.1), and probabilistic programming for dynamic models and inference
(Sec 2.5). It's not clear how to extend Aboleth's design to this broader support without incorporating BL's design.
• **For GPs, Aboleth only supports random feature approximations.** It does not support exact GPs or variational
GPs. For stochastic output layers and reversible layers, Aboleth does not support them. This makes Aboleth primarily
a BNN library. In contrast, BL implements a unifying design across uncertainty-aware functions.
• **Aboleth creates a new neural network language.** BL instead considers how to augment the existing Keras
semantics, unifying deterministic and stochastic functions. (In contrast, for example, `aboleth.DenseVariational`
has a separate design from `aboleth.Dense`.) BL's benefit is building on an existing ecosystem, leveraging the
efficiently optimized deterministic layer computations as well as compatibility across TensorFlow libraries.

**not following up on more recent estimators.** Flipout and Deterministic VI are already available in BL. Implementation-
wise, they subclass, e.g., DenseReparameterization and only reimplement the `call()` method (described in Sec 2.1).

**toolkit is inflexible: "I am under the impression that only VI with Gaussian prior and variational posterior**
**(with diagonal covariance) is supported."** Footnote 2 describes non-Gaussian priors and posteriors. The only
constraint is that the layer initializer returns a (stochastic) Tensor representing a sample; the layer regularizer can utilize
prior/posterior assumptions as necessary. For example, for an implicit prior and posterior, have the initializer simulate
from $q$; have the regularizer compute a density ratio loss involving only the sample and a trainable discriminator.

**"comparison with deterministic in term of training time and memory overhead."** Sec 3.1's deterministic Trans-
former takes 13 hours; the Bayesian Transformer takes 16 hours and 2 extra gb per TPU. Sec 3.2's deterministic
dynamics model takes 20 hours, 8 gb; the Bayesian dynamics model takes 22 hours; 8 gb. We'll add to the revision.

[Meta-Review · NeurIPS 2019]

This work was debated controversially among the reviewers. They all agreed that the work was presented well, and both the idea of the paper and how it is realised as a software interface are novel (or at least a clear improvement over existing frameworks). Software packages generally struggle to get accepted at major conferences. The discussion between the reviewers hinged primarily on this point, too. I would thus like to throw my own vote in for this paper. It is true that the community has not yet developed a good and consistent way to evaluate software contributions, in particular vis-a-vis theoretical and empirical papers. But it is high time that our community becomes more professional in software development. There is an abundance of papers at NeurIPS that provide insufficient or even no implementation details, and this leads to a shocking amount of published work never getting reproduced, used or even widely discussed. As the field matures, we need much more software development in ML. But software design is a thankless, unappreciated task in the academic community. It is time-consuming, and often met with *more* scrutiny by reviewers precisely because it is more accessible, and it's always easy to find something to criticise about a software toolkit. But good software toolkits are among the most impactful works of the past years in this community! (We wouldn't have had the deep learning boom without software toolkits). And not all software can and will be developed by commercial players. There are software tasks that just don't have an obvious business case. Examples include software for benchmarking and comparison of existing methods, and novel general functionality that does not address a specific application domain. I think the present paper fits into this domain, even if it is perhaps not the best example. The community indeed urgently needs to develop fair and transparent standards for how software contributions should be evaluated and (most importantly) appreciated. But it is not the present paper's fault that this has not yet happened, so this problem should not be held against it. It thus recommend that this work should be accepted. Having said all this, I also strongly encourage the authors to clean up the presentation of this paper. Right now, the layout is not clean and at times hard to read. Please take all the criticism of the reviewers into account when preparing the camera-ready version to ensure this paper actually reaches the NeurIPS audience.